# Leader Stochastic Gradient Descent for Distributed Training of Deep Learning Models

**Yunfei Teng**[*,1]
yt1208@nyu.edu

**Wenbo Gao**[*,2]
wg2279@columbia.edu

**Francois Chalus**
chalusf3@gmail.com

**Anna Choromanska**
ac5455@nyu.edu

**Donald Goldfarb**
goldfarb@columbia.edu

**Adrian Weller**
aw665@cam.ac.uk

## Abstract

We consider distributed optimization under communication constraints for training deep learning models. We propose a new algorithm, whose parameter updates rely on two forces: a regular gradient step, and a corrective direction dictated by the currently best-performing worker (leader). Our method differs from the parameter-averaging scheme EASGD [1] in a number of ways: (i) our objective formulation does not change the location of stationary points compared to the original optimization problem; (ii) we avoid convergence decelerations caused by pulling local workers descending to different local minima to each other (i.e. to the average of their parameters); (iii) our update by design breaks the curse of symmetry (the phenomenon of being trapped in poorly generalizing sub-optimal solutions in symmetric non-convex landscapes); and (iv) our approach is more communication efficient since it broadcasts only parameters of the leader rather than all workers. We provide theoretical analysis of the batch version of the proposed algorithm, which we call Leader Gradient Descent (LGD), and its stochastic variant (LSGD). Finally, we implement an asynchronous version of our algorithm and extend it to the multi-leader setting, where we form groups of workers, each represented by its own local leader (the best performer in a group), and update each worker with a corrective direction comprised of two attractive forces: one to the local, and one to the global leader (the best performer among all workers). The multi-leader setting is well-aligned with current hardware architecture, where local workers forming a group lie within a single computational node and different groups correspond to different nodes. For training convolutional neural networks, we empirically demonstrate that our approach compares favorably to state-of-the-art baselines.

## 1 Introduction

As deep learning models and data sets grow in size, it becomes increasingly helpful to parallelize their training over a distributed computational environment. These models lie at the core of many modern machine-learning-based systems for image recognition [2], speech recognition [3], natural language processing [4], and more. This paper focuses on the parallelization of the data, not the model, and considers collective communication scheme [5] that is most commonly used nowadays. A typical approach to data parallelization in deep learning [6, 7] uses multiple workers that run variants of SGD [8] on different data batches. Therefore, the effective batch size is increased by the number of workers. Communication ensures that all models are synchronized and critically relies on a scheme where each worker broadcasts its parameter gradients to all the remaining workers.

This is the case for DOWNPOUR [9] (its decentralized extension, with no central parameter server, based on the ring topology can be found in [10]) or Horovod [11] methods. These techniques require frequent communication (after processing each batch) to avoid instability/divergence, and hence are communication expensive. Moreover, training with a large batch size usually hurts generalization [12, 13, 14] and convergence speed [15, 16].

Another approach, called Elastic Averaging (Stochastic) Gradient Decent, EA(S)GD [1], introduces elastic forces linking the parameters of the local workers with central parameters computed as a moving average over time and space (i.e. over the parameters computed by local workers). This method allows less frequent communication as workers by design do not need to have the same parameters but are instead periodically pulled towards each other. The objective function of EASGD, however, has stationary points which are not stationary points of the underlying objective function (see Proposition 8 in the Supplement), thus optimizing it may lead to sub-optimal solutions for the original problem. Further, EASGD can be viewed as a parallel extension of the averaging SGD scheme [17] and as such it inherits the downsides of the averaging policy. On non-convex problems, when the iterates are converging to different local minima (that may potentially be globally optimal), the averaging term can drag the iterates in the wrong directions and significantly hurt the convergence speed of both local workers and the master. In symmetric regions of the optimization landscape, the elastic forces related with different workers may cancel each other out causing the master to be permanently stuck in between or at the maximum between different minima, and local workers to be stuck at the local minima or on the slopes above them. This can result in arbitrarily bad generalization error. We refer to this phenomenon as the "curse of symmetry". Landscape symmetries are common in a plethora of non-convex problems [18, 19, 20, 21, 22], including deep learning [23, 24, 25, 26].

This paper revisits the EASGD update and modifies it in a simple, yet powerful way which overcomes the above mentioned shortcomings of the original technique. We propose to replace the elastic force relying on the average of the parameters of local workers by an attractive force linking the local workers and the current best performer among them (leader). Our approach reduces the communication overhead related with broadcasting parameters of all workers to each other, and instead requires broadcasting only the leader parameters. The proposed approach easily adapts to a typical hardware architecture comprising of multiple compute nodes where each node contains a group of workers and local communication, within a node, is significantly faster than communication between

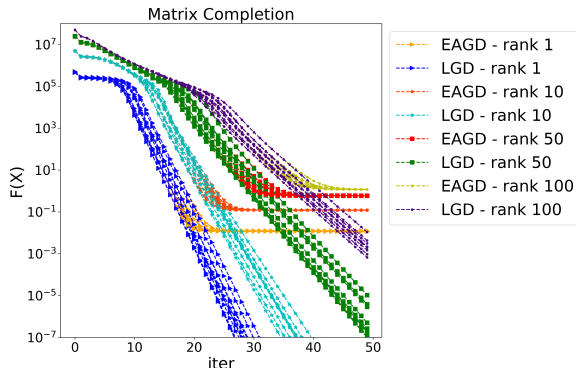

Figure 1: Low-rank matrix completion problems solved with EAGD and LGD. The dimension $d = 1000$ and four ranks $r \in \{1, 10, 50, 100\}$ are used. The reported value for each algorithm is the value of the best worker (8 workers are used in total) at each step.

the nodes. We propose a multi-leader extension of our approach that adapts well to this hardware architecture and relies on forming groups of workers (one per compute node) which are attracted both to their local and global leader. To reduce the communication overhead, the correction force related with the global leader is applied less frequently than the one related with the local leader.

Finally, our L(S)GD approach, similarly to EA(S)GD, tends to explore wide valleys in the optimization landscape when the pulling force between workers and leaders is set to be small. This property often leads to improved generalization performance of the optimizer [27, 28].

The paper is organized as follows: Section 2 introduces the L(S)GD approach, Section 3 provides theoretical analysis, Section 4 contains empirical evaluation, and finally Section 5 concludes the paper. Theoretical proofs and additional theoretical and empirical results are contained in the Supplement.

## 2 Leader (Stochastic) Gradient Descent "L(S)GD" Algorithm

### 2.1 Motivating example

Figure 1 illustrates how elastic averaging can impair convergence. To obtain the figure we applied EAGD (Elastic Averaging Gradient Decent) and LGD to the matrix completion problem of the form: $\min_X \left\{ \frac{1}{4} \|M - XX^T\|_F^2 : X \in \mathbb{R}^{d \times r} \right\}$. This problem is non-convex but is known to have the property that all local minimizers are global minimizers [18]. For four choices of the rank $r$, we generated 10 random instances of the matrix completion problem, and solved each with EAGD and LGD, initialized from the same starting points (we use 8 workers). For each algorithm, we report the progress of the *best* objective value at each iteration, over all workers. Figure 1 shows the results across 10 random experiments for each rank.

It is clear that EAGD slows down significantly as it approaches a minimizer. Typically, the center $\widetilde{X}$ of EAGD is close to the average of the workers, which is a poor solution for the matrix completion problem when the workers are approaching different local minimizers, even though all local minimizers are globally optimal. This induces a pull on each node *away* from the minimizers, which makes it extremely difficult for EAGD to attain a solution of high accuracy. In comparison, LGD does not have this issue. Further details of this experiment, and other illustrative examples of the difference between EAGD and LGD, can be found in the Supplement.

### 2.2 Symmetry-breaking updates

Next we explain the basic update of the L(S)GD algorithm. Consider first the single-leader setting and the problem of minimizing loss function $L$ in a parallel computing environment. The optimization problem is given as

$$\min_{x^1, x^2, \ldots, x^l} L(x^1, x^2, \ldots, x^l) := \min_{x^1, x^2, \ldots, x^l} \sum_{i=1}^{l} \mathbb{E}[f(x^i; \xi^i)] + \frac{\lambda}{2} \|x^i - \tilde{x}\|^2, \qquad (1)$$

where $l$ is the number of workers, $x^1, x^2, \ldots, x^l$ are the parameters of the workers and $\tilde{x}$ are the parameters of the leader. The best performing worker, i.e. $\tilde{x} = \arg\min_{x^1, x^2, \ldots, x^l} \mathbb{E}[f(x^i; \xi^i)]$), and $\xi^i$s are data samples drawn from some probability distribution $\mathcal{P}$. $\lambda$ is the hyperparameter that denotes the strength of the force pulling the workers to the leader. In the theoretical section we will refer to $\mathbb{E}[f(x^i; \xi^i)]$ as simply $f(x^i)$. This formulation can be further extended to the multi-leader setting. The optimization problem is modified to the following form

$$\min_{x^{1,1}, x^{1,2}, \ldots, x^{n,l}} L(x^{1,1}, x^{1,2}, \ldots, x^{n,l})$$

$$:= \min_{x^{1,1}, x^{1,2}, \ldots, x^{n,l}} \sum_{j=1}^{n} \sum_{i=1}^{l} \mathbb{E}[f(x^{j,i}; \xi^{j,i})] + \frac{\lambda}{2} \|x^{j,i} - \tilde{x}^j\|^2 + \frac{\lambda_G}{2} \|x^{j,i} - \tilde{x}\|^2, \qquad (2)$$

where $n$ is the number of groups, $l$ is the number of workers in each group, $\tilde{x}^j$ is the local leader of the $j^{\text{th}}$ group (i.e. $\tilde{x}^j = \arg\min_{x^{j,1}, x^{j,2}, \ldots, x^{j,l}} \mathbb{E}[f(x^{j,i}; \xi^{j,i})]$), $\tilde{x}$ is the global leader (the best worker among local leaders, i.e. $\tilde{x} = \arg\min_{x^{1,1}, x^{1,2}, \ldots, x^{n,l}} \mathbb{E}[f(x^{j,i}; \xi^{j,i})]$), $x^{j,1}, x^{j,2}, \ldots, x^{j,l}$ are the parameters of the workers in the $j^{\text{th}}$ group, and $\xi^{j,i}$s are the data samples drawn from $\mathcal{P}$. $\lambda$ and $\lambda_G$ are the hyperparameters that denote the strength of the forces pulling the workers to their local and global leader respectively.

The updates of the LSGD algorithm are captured below, where $t$ denotes iteration. The first update shown in Equation 3 is obtained by taking the gradient descent step on the objective in Equation 2 with respect to variables $x^{j,i}$. The stochastic gradient of $\mathbb{E}[f(x^i; \xi^i)]$ with respect to $x^{j,i}$ is denoted as $g_t^{j,i}$ (in case of LGD the gradient is computed over all training examples) and $\eta$ is the learning rate.

$$x_{t+1}^{j,i} = x_t^{j,i} - \eta g_t^{j,i}(x_t^{j,i}) - \lambda(x_t^{j,i} - \tilde{x}_t^j) - \lambda_G(x_t^{j,i} - \tilde{x}_t) \qquad (3)$$

where $\tilde{x}_{t+1}^j$ and $\tilde{x}_{t+1}$ are the local and global leaders defined above.

Equation 3 describes the update of any given worker and is comprised of the regular gradient step and two corrective forces (in single-leader setting the third term disappears as $\lambda_G = 0$ then). These

---

**Algorithm 1** LSGD Algorithm (Asynchronous)

---

**Input:** pulling coefficients $\lambda, \lambda_G$, learning rate $\eta$, local/global communication periods $\tau, \tau_G$

**Initialize:**

         Randomly initialize $x^{1,1}, x^{1,2}, ..., x^{n,l}$

         Set iteration counters $t^{j,i} = 0$

         Set $\tilde{x}_0^j = \underset{x^{j,1},...,x^{j,l}}{\arg\min} \mathbb{E}[f(x^{j,i}; \xi_0^{j,i})], \tilde{x}_0 = \underset{x^{1,1},...,x^{n,l}}{\arg\min} \mathbb{E}[f(x^{j,i}; \xi_0^{j,i})];$

**repeat**

     **for all** $j = 1, 2, \ldots, n, i = 1, 2, \ldots, l$ **do**               ▷ Do in parallel for each worker

         Draw random sample $\xi_{t^{j,i}}^{j,i}$

         $x^{j,i} \leftarrow x^{j,i} - \eta g_t^{j,i}(x^{j,i})$

         $t^{j,i} = t^{j,i} + 1;$

         **if** $nl\tau$ divides $(\sum_{j=1}^{n} \sum_{i=1}^{l} t^{j,i})$ **then**

              $\tilde{x}^j = \arg\min_{x^{j,1},...,x^{j,l}} \mathbb{E}[f(x^{j,i}; \xi_{t^{j,i}}^{j,i})].$         ▷ Determine the local best workers

              $x^{j,i} \leftarrow x^{j,i} - \lambda(x^{j,i} - \tilde{x}^j)$               ▷ Pull to the local best workers

         **end if**

         **if** $nl\tau_G$ divides $(\sum_{j=1}^{n} \sum_{i=1}^{l} t^{j,i})$ **then**

              $\tilde{x} = \arg\min_{x^{1,1},...,x^{n,l}} \mathbb{E}[f(x^{j,i}; \xi_{t^{j,i}}^{j,i})].$         ▷ Determine the global best worker

              $x^{j,i} \leftarrow x^{j,i} - \lambda_G(x^{j,i} - \tilde{x})$               ▷ Pull to the global best worker

         **end if**

     **end for**

**until** termination

---

forces constitute the communication mechanism among the workers and pull all the workers towards the currently best local and global solution to ensure fast convergence. As opposed to EASGD, the updates performed by workers in LSGD break the curse of symmetry and avoid convergence decelerations that result from workers being pulled towards the average which is inherently influenced by poorly performing workers. In this paper, instead of pulling workers to their averaged parameters, we propose the mechanism of pulling the workers towards the leaders. The flavor of the update resembles a particle swarm optimization approach [29], which is not typically used in the context of stochastic gradient optimization for deep learning. Our method may therefore be viewed as a dedicated particle swarm optimization approach for training deep learning models in the stochastic setting and parallel computing environment.

Next we describe the LSGD algorithm in more detail. We rely on the collective communication scheme. In order to reduce the amount of communication between the workers, it is desired to pull them towards the leaders less often than every iteration. Also, in practice each worker can have a different speed. To prevent waiting for the slower workers and achieve communication efficiency, we implement the algorithm in the asynchronous operation mode. In this case, the communication period is determined based on the total number of iterations computed across all workers and the communication is performed every $nl\tau$ or $nl\tau_G$ iterations, where $\tau$ and $\tau_G$ denote local and global communication periods, respectively. In practice, we use $\tau_G > \tau$ since communication between workers lying in different groups is more expensive than between workers within one group, as explained above. When communication occurs, all workers are updated at the same time (i.e. pulled towards the leaders) in order to take advantage of the collective communication scheme. Between communications, workers run their own local SGD optimizers. The resulting LSGD method is very simple, and is depicted in Algorithm 1.

The next section provides a theoretical description of the single-leader batch (LGD) and stochastic (LSGD) variants of our approach.

# 3 Theoretical Analysis

We assume without loss of generality that there is a single leader. The objective function with multiple leaders is given by $f(x) + \frac{\lambda_1}{2}\|x - z_1\|^2 + \ldots + \frac{\lambda_c}{2}\|x - z_c\|^2$, which is equivalent to $f(x) + \frac{\Lambda}{2}\|x - \widetilde{z}\|^2$ for $\Lambda = \sum_{i=1}^c \lambda_i$ and $\widetilde{z} = \frac{1}{\Lambda}\sum_{i=1}^c \lambda_i z_i$. Proofs for this section are deferred to the Supplement.

## 3.1 Convergence Rates for Stochastic Strongly Convex Optimization

We first show that LSGD obtains the same convergence rate as SGD for stochastic strongly convex problems [30]. In Section 3.3 we discuss how and when LGD can obtain *better* search directions than gradient descent. We discuss non-convex optimization in Section 3.2. Throughout Section 3.1, $f$ will typically satisfy:

**Assumption 1** $f$ is $M$-Lipschitz-differentiable and $m$-strongly convex, which is to say, the gradient $\nabla f$ satisfies $\|\nabla f(x) - \nabla f(y)\| \le M\|x - y\|$, and $f$ satisfies $f(y) \ge f(x) + \nabla f(x)^T(y - x) + \frac{m}{2}\|y - x\|^2$. We write $x^*$ for the unique minimizer of $f$, and $\kappa := \frac{M}{m}$ for the condition number of $f$.

### 3.1.1 Convergence Rates

The key technical result is that LSGD satisfies a similar one-step descent in expectation as SGD, with an additional term corresponding to the pull of the leader. To provide a unified analysis of 'pure' LSGD as well as more practical variants where the leader is updated infrequently or with errors, we consider a general iteration $x_+ = x - \eta(\widetilde{g}(x) + \lambda(x - z))$, where $z$ is an arbitrary guiding point; that is, $z$ may not be the minimizer of $x^1, \ldots, x^p$, nor even satisfy $f(z) \le f(x^i)$. Since the nodes operate independently except when updating $z$, we may analyze LSGD steps for each node individually, and we write $x = x^i$ for brevity.

**Theorem 1.** *Let $f$ satisfy Assumption 1. Let $\widetilde{g}(x)$ be an unbiased estimator for $\nabla f(x)$ with $\mathrm{Var}(\widetilde{g}(x)) \le \sigma^2 + \nu\|\nabla f(x)\|^2$, and let $z$ be any point. Suppose that $\eta, \lambda$ satisfy $\eta \le (2M(\nu + 1))^{-1}$ and $\eta\lambda \le (2\kappa)^{-1}, \eta\sqrt{\lambda} \le (\kappa\sqrt{2m})^{-1}$. Then the LSGD step satisfies*

$$\mathbb{E}f(x_+) - f(x^*) \le (1 - m\eta)(f(x) - f(x^*)) - \eta\lambda(f(x) - f(z)) + \frac{\eta^2 M}{2}\sigma^2. \tag{4}$$

*Note the presence of the new term $-\eta\lambda(f(x) - f(z))$ which speeds up convergence when $f(z) \le f(x)$, i.e the leader is better than $x$. If the leader $z_k$ is always chosen so that $f(z_k) \le f(x_k)$ at every step $k$, then $\limsup_{k\to\infty} \mathbb{E}f(x_k) - f(x^*) \le \frac{1}{2}\eta\kappa\sigma^2$. If $\eta$ decreases at the rate $\eta_k = \Theta(\frac{1}{k})$, then $\mathbb{E}f(x_k) - f(x^*) \le O(\frac{1}{k})$.*

The $O(\frac{1}{k})$ rate of LSGD matches that of comparable distributed methods. Both Hogwild [31] and EASGD achieve a rate of $O(\frac{1}{k})$ on strongly convex objective functions. We note that published convergence rates are not available for many distributed algorithms (including DOWNPOUR [9]).

### 3.1.2 Communication Periods

In practice, communication between distributed machines is costly. The LSGD algorithm has a *communication period* $\tau$ for which the leader is only updated every $\tau$ iterations, so each node can run independently during that period. This $\tau$ is allowed to differ between nodes, and over time, which captures the asynchronous and multi-leader variants of LSGD. We write $x_{k,j}$ for the $j$-th step during the $k$-th period. It may occur that $f(z) > f(x_{k,j})$ for some $k, j$, that is, the current solution $x_{k,j}$ is now better than the last selected leader. In this case, the leader term $\lambda(x - z)$ may no longer be beneficial, and instead simply pulls $x$ toward $z$. There is no general way to determine how many steps are taken before this event. However, we can show that if $f(z) \ge f(x)$, then

$$\mathbb{E}f(x_+) \le f(z) + \frac{1}{2}\eta^2 M\sigma^2, \tag{5}$$

so the solution will not become *worse* than a stale leader (up to gradient noise). As $\tau$ goes to infinity, LSGD converges to the minimizer of $\psi(x) = f(x) + \frac{\lambda}{2}\|x - z\|^2$, which is quantifiably better than $z$ as captured in Theorem 2. Together, these facts show that LSGD is safe to use with long communication periods as long as the original leader is good.

**Theorem 2.** *Let $f$ be $m$-strongly convex, and let $x^*$ be the minimizer of $f$. For fixed $\lambda, z$, define $\psi(x) = f(x) + \frac{\lambda}{2}\|x - z\|^2$. The minimizer $w$ of $\psi$ satisfies $f(w) - f(x^*) \leq \frac{\lambda}{m+\lambda}(f(z) - f(x^*))$.*

The theoretical results here and in Section 3.1.1 address two fundamental instances of the LSGD algorithm: the 'synchronous' case where communication occurs each round, and the 'infinitely asynchronous' case where communication periods are arbitrarily long. For unknown periods $\tau > 1$, it is difficult to demonstrate general quantifiable improvements beyond (5), but we note that (4), Theorem 2, and the results on stochastic leader selection (Sections 3.1.3 and 7.6) can be combined to analyze specific instances of the asynchronous LSGD.

In our experiments, we employ another method to avoid the issue of stale leaders. To ensure that the leader is good, we perform an LSGD step only on the first step after a leader update, and then take standard SGD steps for the remainder of the communication period.

### 3.1.3 Stochastic Leader Selection

Next, we consider the impact of selecting the leader with errors. In practice, it is often costly to evaluate $f(x)$, as in deep learning. Instead, we estimate the values $f(x^i)$, and then select $z$ as the variable having the smallest estimate. Formally, suppose that we have an unbiased estimator $\widetilde{f}(x)$ of $f(x)$, with uniformly bounded variance. At each step, a single sample $y_1, \ldots, y_p$ is drawn from each estimator $\widetilde{f}(x^1), \ldots, \widetilde{f}(x^p)$, and then $z = \{x^i : y_i = \min\{y_1, \ldots, y_p\}\}$. We refer to this as *stochastic leader selection*. The stochastic leader satisfies $\mathbb{E}f(z) \leq f(z_{true}) + 4\sqrt{p}\sigma_f$, where $z_{true}$ is the true leader (see supplementary materials). Thus, the error introduced by the stochastic leader contributes an additive error of at most $4\eta\lambda\sqrt{p}\sigma_f$. Since this is of order $\eta$ rather than $\eta^2$, we cannot guarantee convergence with $\eta_k = \Theta(\frac{1}{k})$[1] unless $\lambda_k$ is also decreasing. We have the following result:

**Theorem 3.** *Let $f$ satisfy Assumption 1, and let $\widetilde{g}(x)$ be as in Theorem 1. Suppose we use stochastic leader selection with $\widetilde{f}(x)$ having $\mathrm{Var}(\widetilde{f}(x)) \leq \sigma_f^2$. If $\eta, \lambda$ are fixed so that $\eta \leq (2M(\nu + 1))^{-1}$ and $\eta\lambda \leq (2\kappa)^{-1}, \eta\sqrt{\lambda} \leq (\kappa\sqrt{2m})^{-1}$, then $\limsup_{k\to\infty} \mathbb{E}f(x_k) - f(x^*) \leq \frac{1}{2}\eta\kappa\sigma^2 + \frac{4}{m}\lambda\sqrt{p}\sigma_f$. If $\eta, \lambda$ decrease at the rate $\eta_k = \Theta(\frac{1}{k}), \lambda_k = \Theta(\frac{1}{k})$, then $\mathbb{E}f(x_k) - f(x^*) \leq O(\frac{1}{k})$.*

The communication period and the accuracy of stochastic leader selection are both methods of reducing the cost of updating the leader, and can be substitutes. When the communication period is long, it may be effective to estimate $f(x^i)$ to higher accuracy, since this can be done independently.

### 3.2 Non-convex Optimization: Stationary Points

As mentioned above, EASGD has the flaw that the EASGD objective function can have stationary points such that none of $x^1, \ldots, x^p, \widetilde{x}$ is a stationary point of the underlying function $f$. LSGD does not have this issue.

**Theorem 4.** *Let $\Omega_i$ be the points $(x^1, \ldots, x^p)$ where $x^i$ is the unique minimizer among $(x^1, \ldots, x^p)$. If $x^* = (w^1, \ldots, w^p) \in \Omega_i$ is a stationary point of the LSGD objective function, then $\nabla f^i(w^i) = 0$.*

Moreover, it can be shown that for the deterministic algorithm LGD with *any choice of communication periods*, there will always be some variable $x^i$ such that $\liminf \|\nabla f(x_k^i)\| = 0$.

**Theorem 5.** *Assume that $f$ is bounded below and $M$-Lipschitz-differentiable, and that the LGD step sizes are selected so that $\eta_i < \frac{2}{M}$. Then for any choice of communication periods, it holds that for every $i$ such that $x^i$ is the leader infinitely often, $\liminf_k \|\nabla f(x_k^i)\| = 0$.*

### 3.3 Search Direction Improvement from Leader Selection

In this section, we discuss how LGD can obtain better search directions than gradient descent. In general, it is difficult to determine when the LGD step will satisfy $f(x - \eta(\nabla f(x) + \lambda(x - z))) \leq f(x - \eta\nabla f(x))$, since this depends on the precise combination of $f, x, z, \eta, \lambda$, and moreover, the maximum allowable value of $\eta$ is different for LGD and gradient descent. Instead, we measure the goodness of a search direction by the angle it forms with the Newton direction $d_N(x) = -(\nabla^2 f(x))^{-1}\nabla f(x)$. The Newton method is locally quadratically convergent around local minimizers with non-singular

Hessian, and converges in a single step for quadratic functions if $\eta = 1$. Hence, we consider it desirable to have search directions that are close to $d_N$. Let $\theta(u, v)$ denote the angle between $u, v$. Let $d_z = -(\nabla f(x) + \lambda(x-z))$ be the LGD direction with leader $z$, and $d_G(x) = -\nabla f(x)$. The *angle improvement set* is the set of leaders $I_\theta(x, \lambda) = \{z : f(z) \leq f(x), \theta(d_z, d_N(x)) \leq \theta(d_G(x), d_N(x))\}$. The set of candidate leaders is $E = \{z : f(z) \leq f(x)\}$. We aim to show that a large subset of leaders in $E$ belong to $I_\theta(x, \lambda)$.

In this section, we consider the positive definite quadratic $f(x) = \frac{1}{2}x^T A x$ with condition number $\kappa$ and $d_G(x) = -Ax, d_N(x) = -x$. The first result shows that as $\lambda$ becomes sufficiently small, at least half of $E$ improves the angle. We use the $n$-dimensional volume $\mathrm{Vol}(\cdot)$ to measure the relative size of sets: an ellipsoid $E$ given by $E = \{x : x^T A x \leq 1\}$ has volume $\mathrm{Vol}(E) = \det(A)^{-1/2}\mathrm{Vol}(S_n)$, where $S_n$ is the unit ball.

**Theorem 6.** *Let $x$ be any point such that $\theta_x = \theta(d_G(x), d_N(x)) > 0$, and let $E = \{z : f(z) \leq f(x)\}$. Then $\lim_{\lambda \to 0} \mathrm{Vol}(I_\theta(x, \lambda)) \geq \frac{1}{2}\mathrm{Vol}(E)^2$.*

Next, we consider when $\lambda$ is large. We show that points with large angle between $d_G(x), d_N(x)$ exist, which are most suitable for improvement by LGD. For $r \geq 2$, define $S_r = \{x : \cos(\theta(d_G(x), d_N(x))) = \frac{r}{\sqrt{\kappa}}\}$. It can be shown that $S_r$ is nonempty for all $r \geq 2$. We show that for $x \in S_r$ for a certain range of $r$, $I_\theta(x, \lambda)$ is at least half of $E$ *for any choice of $\lambda$.*

**Theorem 7.** *Let $R_\kappa = \{r : \frac{r}{\sqrt{\kappa}} + \frac{r^{3/2}}{\kappa^{1/4}} \leq 1\}$. If $x \in S_r$ for $r \in R_\kappa$, then for any $\lambda \geq 0$, $\mathrm{Vol}(I_\theta(x, \lambda)) \geq \frac{1}{2}\mathrm{Vol}(E)$.*

Note that Theorems 6 and 7 apply only to *convex* functions, or in the neighborhoods of local minimizers where the objective function is locally convex. In nonconvex landscapes, the Newton direction may point towards saddle points [32], which is undesirable; however, since Theorems 6 and 7 do not apply in this situation, these results do not imply that LSGD has harmful behavior. For nonconvex problems, our intuition is that many candidate leaders lie in directions of *negative curvature*, which would actually lead away from saddle points, but this is significantly harder to analyze since the set of candidates is unbounded a priori.

## 4 Experimental Results

### 4.1 Experimental setup

In this section we compare the performance of LSGD with state-of-the-art methods for parallel training of deep networks, such as EASGD and DOWNPOUR (their pseudo-codes can be found in [1]), as well as sequential technique SGD. The codes for LSGD can be found at https://github.com/yunfei-teng/LSGD. We use communication period equal to 1 for DOWNPOUR in all our experiments as this is the typical setting used for this method ensuring stable convergence. The experiments were performed using the CIFAR-10 data set [33] on three benchmark architectures: 7-layer CNN used in the original EASGD paper

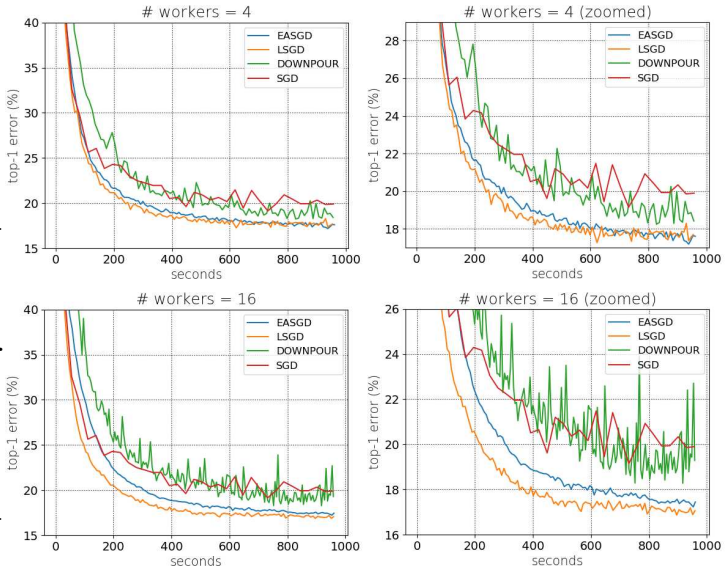

Figure 2: CNN7 on CIFAR-10. Test error for the center variable versus wall-clock time (original plot on the left and zoomed on the right). Test loss is reported in Figure 10 in the Supplement.

(see Section 5.1. in [1]) that we refer to as CNN7, VGG16 [34], and ResNet20 [35]; and ImageNet (ILSVRC 2012) data set [36] on ResNet50.

During training, we select the leader for the LSGD method based on the average of the training loss computed over the last 10 (CIFAR-10) and 64 (ImageNet) data batches. At testing, we report the performance of the center variable for EASGD and LSGD, where for LSGD the center variable is computed as the average of the parameters of all workers. [*Remark*: Note that we use the leader's parameter to pull

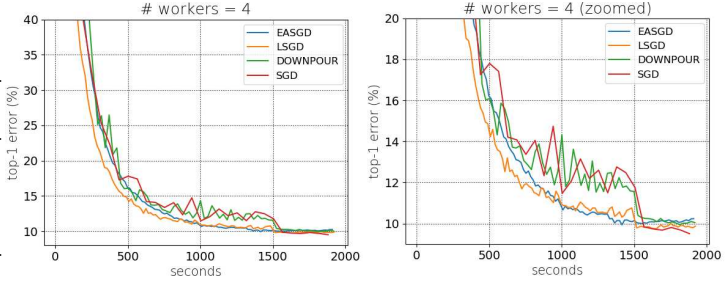

Figure 3: VGG16 on CIFAR-10. Test error for the center variable versus wall-clock time (original plot on the left and zoomed on the right). Test loss is reported in Figure 12 in the Supplement.

to at training and we report the averaged parameters at testing deliberately. It is demonstrated in our paper (e.g.: Figure 1) that pulling workers to the averaged parameters at training may slow down convergence and we address this problem. Note that after training, the parameters that workers obtained after convergence will likely lie in the same valley of the landscape (see [37]) and thus their average is expected to have better generalization ability (e.g. [27, 38]), which is why we report the results for averaged parameters at testing.] Finally, for all methods we use weight decay with decay coefficient set to $10^{-4}$. In our experiments we use either 4 workers (single-leader LSGD setting) or 16 workers (multi-leader LSGD setting with 4 groups of workers). For all methods, we report the learning rate leading to the smallest achievable test error under similar convergence rates (we rejected small learning rates which led to unreasonably slow convergence).

We use GPU nodes interconnected with Ethernet. Each GPU node has four GTX 1080 GPU processors where each local worker corresponds to one GPU processor. We use CUDA Toolkit 10.0[3] and NCCL 2[4]. We have developed a software package based on PyTorch for distributed training, which will be released (details are elaborated in Section 9.4).

Data processing and prefetching are discussed in the Supplement. The summary of the hyperparameters explored for each method are also provided in the Supplement. We use constant learning rate for CNN7 and learning rate drop (we divide the learning rate by 10 when we observe saturation of the optimizer) for VGG16, ResNet20, and ResNet50.

## 4.2 Experimental Results

In Figure 2 we report results obtained with CNN7 on CIFAR-10. We run EASGD and LSGD with communication period $\tau = 64$. We used $\tau_G = 128$ for the multi-leader LSGD case. The number of workers was set to $l = \{4, 16\}$. Our method consistently outperforms the competitors in terms of convergence speed (it is roughly 1.5 times faster than EASGD for 16 workers) and for 16 workers it obtains smaller error.

In Figure 3 we demonstrate results for VGG16 and CIFAR-10 with communication period 64 and number of workers equal to 4. LSGD converges marginally faster than EASGD and recovers

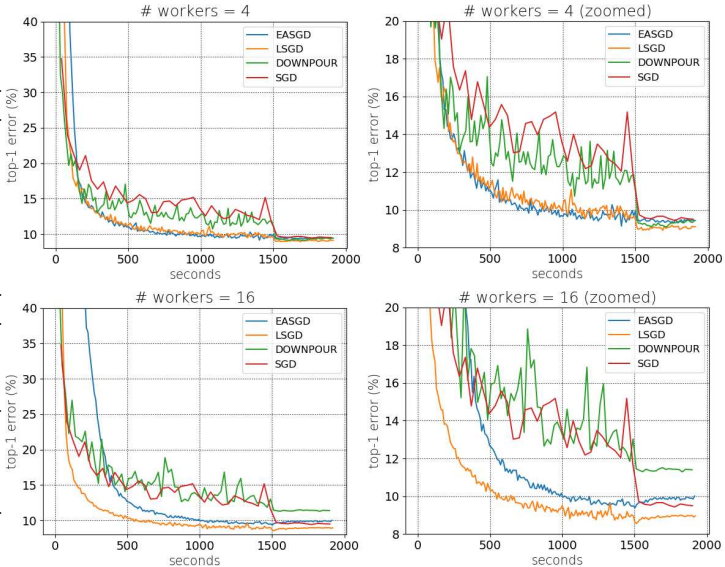

Figure 4: ResNet20 on CIFAR-10. Test error for the center variable versus wall-clock time (original plot on the left and zoomed on the right). Test loss is reported in Figure 11 in the Supplement.

[3]https://developer.nvidia.com/cuda-zone
[4]https://developer.nvidia.com/nccl

the same error. At the same time it outperforms significantly DOWNPOUR in terms of convergence speed and obtains a slightly better solution.

The experimental results obtained using ResNet20 and CIFAR-10 for the same setting of communication period and number of workers as in case of CNN7 are shown in Figure 4. On 4 workers we converge comparably fast to EASGD but recover better test error. For this experiment in Figure 5 we show the switching pattern between the leaders indicating that LSGD indeed takes advantage of all workers when exploring the landscape. On 16 workers we converge roughly 2 times faster than EASGD and obtain significantly smaller error. In this and CNN7 experiment LSGD (as well as EASGD) are consistently better than DONWPOUR and SGD, as expected.

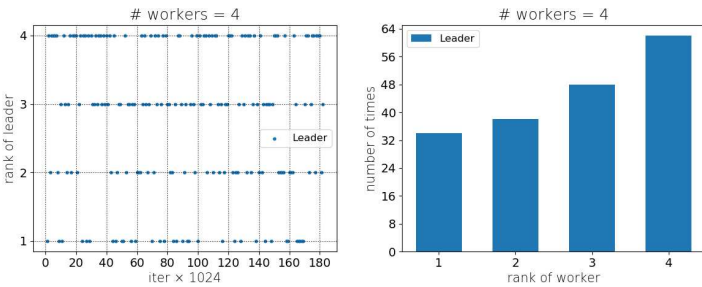

Figure 5: ResNet20 on CIFAR-10. The identity of the worker that is recognized as the leader (i.e. rank) versus iterations (on the left) and the number of times each worker was the leader (on the right).

**Remark 1.** *We believe that these two facts together — (1) the schedule of leader switching recorded in the experiments shows frequent switching, and (2) the leader point itself is not pulled away from minima — suggest that the 'pulling away' in LSGD is beneficial: non-leader workers that were pulled away from local minima later became the leader, and thus likely obtained an even better solution than they originally would have.*

Finally, in Figure 6 we report the empirical results for ResNet50 run on ImageNet. The number of workers was set to 4 and the communication period $\tau$ was set to 64. In this experiment our algorithm behaves comparably to EASGD but converges much faster than DOWNPOUR. Also note that for ResNet50 on ImageNet, SGD is consistently worse than all reported methods (training on ImageNet with SGD on a single GTX1080 GPU until

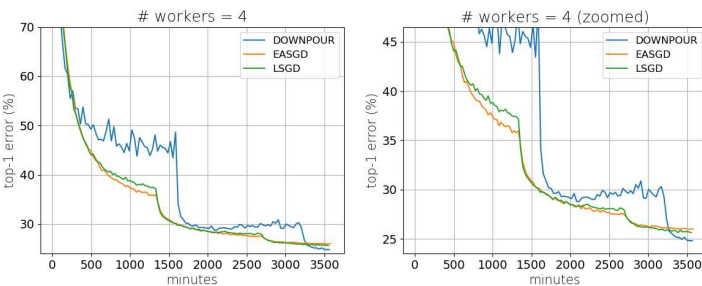

Figure 6: ResNet50 on ImageNet. Test error for the center variable versus wall-clock time (original plot on the left and zoomed on the right). Test loss is reported in Figure 13 in the Supplement.

convergence usually takes about a week and gives slightly worse final performance), which is why the SGD curve was deliberately omitted (other methods converge in around two days).

## 5 Conclusion

In this paper we propose a new algorithm called LSGD for distributed optimization in non-convex settings. Our approach relies on pulling workers to the current best performer among them, rather than their average, at each iteration. We justify replacing the average by the leader both theoretically and through empirical demonstrations. We provide a thorough theoretical analysis, including proof of convergence, of our algorithm. Finally, we apply our approach to the matrix completion problem and training deep learning models and demonstrate that it is well-suited to these learning settings.

## Acknowledgements

WG and DG were supported in part by NSF Grant IIS-1838061. AW acknowledges support from the David MacKay Newton research fellowship at Darwin College, The Alan Turing Institute under EPSRC grant EP/N510129/1 & TU/B/000074, and the Leverhulme Trust via the CFI.

## Footnotes

*,1: Equal contribution. Algorithm development and implementation on deep models.

*,2: Equal contribution. Theoretical analysis and implementation on matrix completion.

[1]For intuition, note that $\sum_{n=1}^{\infty} \frac{1}{n}$ is divergent.

[2]Note that $I_\theta(x, \lambda_1) \supseteq I_\theta(x, \lambda_2)$ for $\lambda_1 \leq \lambda_2$, so the limit is well-defined.

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
