[Supplementary Material]

# Leader Stochastic Gradient Descent for Distributed Training of Deep Learning Models (Supplementary Material)

## Abstract

This Supplement presents additional details in support of the full article. These include the proofs of the theoretical statements from the main body of the paper and additional theoretical results. We also provide a toy illustrative example of the difference between LSGD and EASGD. Finally, the Supplement contains detailed description of the experimental setup and additional experiments and figures to provide further empirical support for the proposed methodology.

## 6 LGD versus EAGD: Illustrative Example

Figure 7: **Left**: Trajectories of variables $(\mathbf{x}, \mathbf{y})$ during optimization. The dashed lines represent the local minima. The red and blue circles are the start and end points of each trajectory, respectively. **Right**: The value of the objective function $L(x, y)$ for each worker during training.

We consider the following non-convex optimization problem:

$$\min_{x,y} L(x, y), \quad \text{where} \quad L(x, y) = \frac{\sin(\sqrt{x^2 + y^2} \cdot \pi)}{\sqrt{x^2 + y^2} \cdot \pi}.$$

Both methods use 4 workers with initial points $(-6, -4)$, $(-15, -18)$, $(20, 11)$ and $(17, 8)$. The communication period is set to 1. The learning rate for both EAGD and LGD equals 0.1. Furthermore, EAGD uses $\beta = 0.43$ and LGD uses $\lambda = 0.1$.

Table 1 captures optima obtained by different methods.

| Optimizer | $L(x, y)$ |
|-----------|-----------|
| EAGD | -0.0912 |
| LGD | **-0.2172** |

Table 1: Optimum $L(x^*, y^*)$ recovered by EAGD and LGD.

Figure 7 captures the optimization trajectories of EAGD and LGD algorithms. Clearly, EAGD suffers from the averaging policy, whereas LGD is able to recover a solution close to the *global optimum*.

# 7 Proofs of Theoretical Results

We provide omitted proofs from the main text.

## 7.1 Definitions and Notation

Recall that the objective function of Leader (Stochastic) Gradient Descent (L(S)GD) is defined as

$$\min_{x^1,\ldots,x^p} L(x^1,\ldots,x^p) := \sum_{i=1}^{p} f(x^i) + \frac{\lambda}{2}\|x^i - \widetilde{x}\|^2 \tag{6}$$

where $\widetilde{x} = \arg\min\{f(x^1),\ldots,f(x^p)\}$. An L(S)GD step is a (stochastic) gradient step applied to $L$. Writing $z = \widetilde{x}$ at a particular $(x^1,\ldots,x^n)$, the update in the variable $x^i$ is

$$x_+^i = x^i - \eta(\nabla f(x^i) + \lambda(x^i - z))$$

Observe that this reduces to a (S)GD step for the variable which is the leader.

Practical variants of the algorithm do not communicate the updated leader at every iteration. Thus, in our analysis, we will generally take $z$ to be an arbitrary guiding point, which is not necessarily the minimizer of $x^1,\ldots,x^p$, nor even satisfy $f(z) \leq f(x^i)$ for all $i$. The required properties of $z$ will be specified on a result-by-result basis.

When discussing the optimization landscape of LSGD, the term 'LSGD objective function' will refer to (6) with $\widetilde{x}$ defined as the argmin.

*Communication periods* are sequences of steps where the leader is not updated. We introduce the notation $x_{k,j}$ for the $j$-th step in the $k$-th period, where the leader $z$ is updated only at the beginning of each period. We write $b_i(k)$ for the number of steps that $x^i$ takes during the $k$-th period. The standard LSGD defined above has $b_i(k) = 1$ for all $i, k$, in which case $x_{k,1}^i = x_k^i$. In addition, let $\widetilde{x}_k = \operatorname{argmin}\{f(x_{k,1}^1),\ldots,f(x_{k,1}^p)\}$, the leader for the $k$-th period.

## 7.2 Stationary Points of EASGD

The EASGD [1] objective function is defined as

$$\min_{x^1,\ldots,x^p,\widetilde{x}} L(x^1,\ldots,x^p,\widetilde{x}) := \sum_{i=1}^{p} f(x^i) + \frac{\lambda}{2}\|x^i - \widetilde{x}\|^2. \tag{7}$$

Observe that unlike LSGD, $\widetilde{x}$ is a decision variable of EASGD. A stationary point of EASGD is a point such that $\nabla L(x^1,\ldots,x^p,\widetilde{x}) = 0$.

**Proposition 8.** *There exists a Lipschitz differentiable function* $f : \mathbb{R} \to \mathbb{R}$ *such that for every* $0 < \lambda \leq 1$, *there exists a point* $(x_\lambda, y_\lambda, 0)$ *which is a stationary point of EASGD with parameter* $\lambda$, *but none of* $\{x_\lambda, y_\lambda, 0\}$ *is a stationary point of* $f$.

*Proof.* Define $f(x)$ by

$$f(x) = \begin{cases} e^{x+1} & \text{if } x < -1 \\ p(x) & \text{if } -1 \le x \le 1 \\ e^{-x+1} & \text{if } x > 1 \end{cases}$$

where $p(x) = a_6 x^6 + \ldots + a_1 x + a_0$ is a sixth-degree polynomial. For $f$ to be Lipschitz differentiable, we will select $p(x)$ to make $f$ twice continuously differentiable, with bounded second derivative. To make $f$ twice continuously differentiable, we must have $p(1) = 1, p'(1) = -1, p''(1) = 1$ and $p(-1) = -1, p'(-1) = 1, p''(-1) = -1$. Since we aim to have $f'(0) \ne 0$, we also will require $f'(0) = p'(0) = 1$. The existence of $p$ is equivalent to the solvability of a linear system, which is easily checked to be invertible. Thus, we deduce that such a function $f$ exists.

It remains to show that for any $0 < \lambda \le 1$, there exists a stationary point $(x, y, 0)$ of EASGD. Set $x = -y$. The first-order condition yields $f'(x) + \lambda x = 0$. Since $\lambda \le 1$, we have $\lambda(1) + f'(1) \le 0$. For $x \ge 1$, $f'(x) = -e^{-x+1}$ is an increasing function, so $f'(x) + \lambda x$ is increasing, and we deduce that there exists a solution $y_\lambda \ge 1$ with $\lambda y_\lambda + f'(y_\lambda) = 0$. By symmetry, $-y_\lambda \le -1$ satisfies $f'(-y_\lambda) + \lambda(-y_\lambda) = 0$, since $f'(x) = e^{x+1}$ for $x \le -1$. Hence, $(-y_\lambda, y_\lambda, 0)$ is a stationary point of EASGD, but none of $\{-y_\lambda, y_\lambda, 0\}$ are stationary points of $f$. $\square$

### 7.3 Technical Preliminaries

Recall the statement of Assumption 1:

**Assumption 1** $f$ is $M$-Lipschitz-differentiable and $m$-strongly convex, which is to say, the gradient $\nabla f$ satisfies $\|\nabla f(x) - \nabla f(y)\| \le M\|x - y\|$, and $f$ satisfies

$$f(y) \ge f(x) + \nabla f(x)^T (y - x) + \frac{m}{2}\|y - x\|^2.$$

We write $x^*$ for the unique minimizer of $f$, and $\kappa := \frac{M}{m}$ for the condition number of $f$.

We will frequently use the following standard result.

**Lemma 9.** *If $f$ is $M$-Lipschitz-differentiable, then*

$$f(y) \le f(x) + \nabla f(x)^T (y - x) + \frac{M}{2}\|y - x\|^2.$$

*Proof.* See [30, eq. (4.3)]. $\square$

**Lemma 10.** *Let $f$ be $m$-strongly convex, and let $x^*$ be the minimizer of $f$. Then*

$$f(w) - f(x^*) \le \frac{1}{2m}\|\nabla f(w)\|^2 \tag{8}$$

*and*

$$f(w) - f(x^*) \ge \frac{m}{2}\|w - x^*\|^2 \tag{9}$$

*Proof.* Equation (8) is the well-known Polyak-Łojasiewicz inequality. Equation (9) follows from the definition of strong convexity, and $\nabla f(x^*) = 0$. $\square$

**Lemma 11.** *Let $f$ be $M$-Lipschitz-differentiable. If the gradient descent step size $\eta < \frac{2}{M}$, then $\|\nabla f(x)\|^2 \le \alpha(f(x) - f(x^+))$, where $\alpha = \frac{2}{\eta(2 - \eta M)}$.*

*Proof.* By Theorem 9,

$$f(x_+) \le f(x) - \eta\|\nabla f(x)\|^2 + \frac{\eta^2}{2}M\|\nabla f(x)\|^2$$

$$= f(x) - \frac{\eta}{2}(2 - \eta M)\|\nabla f(x)\|^2$$

Rearranging yields the desired result. $\square$

## 7.4 Proofs from Section 3.1.1

**Lemma 12** (One-Step Descent). *Let $f$ satisfy Assumption 1. Let $\widetilde{g}(x)$ be an unbiased estimator for $\nabla f(x)$ with $\mathrm{Var}(\widetilde{g}(x)) \leq \sigma^2 + \nu\|\nabla f(x)\|^2$. Let $x$ be the current iterate, and let $z$ be another point, with $\delta := x - z$. The LSGD step $x_+ = x - \eta(\widetilde{g}(x) + \lambda(x - z))$ satisfies:*

$$\mathbb{E}f(x_+) \leq f(x) - \frac{\eta}{2}(1 - \eta M(\nu + 1))\|\nabla f(x)\|^2 - \frac{\eta}{4}\lambda(m - 2\eta M\lambda)\|\delta\|^2 \tag{10}$$

$$- \frac{\eta\sqrt{\lambda}}{\sqrt{2}}(\sqrt{m} - \eta M\sqrt{2\lambda})\|\nabla f(x)\|\|\delta\| - \eta\lambda(f(x) - f(z)) + \frac{\eta^2}{2}M\sigma^2$$

*where the expectation is with respect to $\widetilde{g}(x)$, and conditioned on the current point $x$. Hence, for sufficiently small $\eta, \lambda$ with $\eta \leq (2M(\nu + 1))^{-1}$ and $\eta\lambda \leq (2\kappa)^{-1}, \eta\sqrt{\lambda} \leq (\kappa\sqrt{2m})^{-1}$,*

$$\mathbb{E}f(x_+) - f(x^*) \leq (1 - m\eta)(f(x) - f(x^*)) - \eta\lambda(f(x) - f(z)) + \frac{\eta^2 M}{2}\sigma^2 \tag{11}$$

*Proof.* The proof is similar to the convergence analysis of SGD. We apply Theorem 9 to obtain

$$f(x_+) \leq f(x) - \eta\nabla f(x)^T(\widetilde{g}(x) + \lambda\delta) + \frac{\eta^2}{2}M\|\widetilde{g}(x) + \lambda\delta\|^2.$$

Taking the expectation and using $\mathbb{E}\widetilde{g}(x) = \nabla f(x)$,

$$\mathbb{E}f(x_+) \leq f(x) - \eta\|\nabla f(x)\|^2 - \eta\lambda\nabla f(x)^T\delta + \frac{\eta^2\lambda^2}{2}M\|\delta\|^2 + \eta^2\lambda M\nabla f(x)^T\delta + \frac{\eta^2}{2}M\mathbb{E}[\widetilde{g}(x)^T\widetilde{g}(x)]$$

Using the definition of $m$-strong convexity, we have $f(z) \geq f(x) - \nabla f(x)^T\delta + \frac{m}{2}\|\delta\|^2$, from which we deduce that $-\nabla f(x)^T\delta \leq -(f(x) - f(z) + \frac{m}{2}\|\delta\|^2)$. Substituting this above, and splitting both the terms $\eta\|\nabla f(x)\|^2, \frac{\eta}{2}m\lambda\|\delta\|^2$ in half, we obtain

$$\mathbb{E}f(x_+) = f(x) - \frac{\eta}{2}\|\nabla f(x)\|^2 + \frac{\eta^2}{2}M\mathbb{E}[\widetilde{g}(x)^T\widetilde{g}(x)]$$

$$- \frac{\eta}{4}m\lambda\|\delta\|^2 + \frac{\eta^2}{2}\lambda^2 M\|\delta\|^2$$

$$- \frac{\eta}{2}\|\nabla f(x)\|^2 - \frac{\eta}{4}m\lambda\|\delta\|^2 + \eta^2\lambda M\nabla f(x)^T\delta$$

$$- \eta\lambda(f(x) - f(z))$$

We proceed to bound each line. For the first line, the standard bias-variance decomposition yields

$$\mathbb{E}[\widetilde{g}(x)^T\widetilde{g}(x)] \leq (\nu + 1)\|\nabla f(x)\|^2 + \sigma^2$$

and so we have

$$-\frac{\eta}{2}\|\nabla f(x)\|^2 + \frac{\eta^2}{2}M\mathbb{E}[\widetilde{g}(x)^T\widetilde{g}(x)] \leq -\frac{\eta}{2}(1 - \eta M(\nu + 1))\|\nabla f(x)\|^2 + \frac{\eta^2}{2}M\sigma^2.$$

For the second line, we obtain

$$-\frac{\eta}{4}m\lambda\|\delta\|^2 + \frac{\eta^2}{2}\lambda^2 M\|\delta\|^2 \leq -\frac{\eta}{4}\lambda(m - 2\eta M\lambda)\|\delta\|^2.$$

For the third line, we apply the inequality $a^2 + b^2 \geq 2ab$ to obtain

$$\frac{\eta}{2}\|\nabla f(x)\|^2 + \frac{\eta}{4}m\lambda\|\delta\|^2 \geq \frac{\eta}{\sqrt{2}}\sqrt{m\lambda}\|\nabla f(x)\|\|\delta\|.$$

Using the Cauchy-Schwarz inequality, we then obtain

$$-\frac{\eta}{2}\|\nabla f(x)\|^2 - \frac{\eta}{4}m\lambda\|\delta\|^2 + \eta^2\lambda\nabla M f(x)^T\delta \leq -\frac{\eta\sqrt{\lambda}}{\sqrt{2}}(\sqrt{m} - \eta M\sqrt{2\lambda})\|\nabla f(x)\|\|\delta\|.$$

Combining these inequalities yields the desired result. $\qquad\square$

**Theorem 13.** *Let $f$ satisfy Assumption 1. Suppose that the leader $z_k$ is always chosen so that $f(z_k) \leq f(x_k)$. If $\eta, \lambda$ are fixed so that $\eta \leq (2M(\nu+1))^{-1}$ and $\eta\lambda \leq (2\kappa)^{-1}, \eta\sqrt{\lambda} \leq (\kappa\sqrt{2m})^{-1}$, then $\limsup_{k\to\infty} \mathbb{E}f(x_k) - f(x^*) \leq \frac{1}{2}\eta\kappa\sigma^2$. If $\eta$ decreases at the rate $\eta_k = \Theta(\frac{1}{k})$, then $\mathbb{E}f(x_k) - f(x^*) = O(\frac{1}{k})$.*

*Proof.* This result follows (11) and Theorems 4.6 and 4.7 of [30]. □

### 7.5 Proofs from Section 3.1.2

**Theorem 14.** *Let $f$ satisfy Assumption 1. Suppose that $\eta, \lambda$ are small enough that $\eta\lambda \leq 1$ and $\eta \leq (2M(\nu+1))^{-1}, \eta\lambda \leq (2\kappa)^{-1}, \eta\sqrt{\lambda} \leq (\kappa\sqrt{2m})^{-1}$. If $f(x) \leq f(z)$, then $\mathbb{E}f(x_+) \leq f(z) + \frac{1}{2}\eta^2 M\sigma^2$.*

*Proof.* This follows from (13), by combining $f(x) - \eta\lambda(f(x) - f(z))$, and using $f(z) \geq f(x)$. □

**Theorem 15.** *Let $f$ be $m$-strongly convex, and let $x^*$ be the minimizer of $f$. Fix a constant $\lambda$ and any point $z$, and define the function $\psi(x) = f(x) + \frac{\lambda}{2}\|x - z\|^2$. Since $\psi$ is strongly convex, it has a unique minimizer $w$. The minimizer $w$ satisfies*

$$f(w) - f(x^*) \leq \frac{\lambda}{m + \lambda}(f(z) - f(x^*)) \tag{12}$$

*and[5]*

$$\|w - x^*\|^2 \leq \frac{\lambda^2}{m(m+\lambda)}\|z - x^*\|^2 \tag{13}$$

*Proof.* The first-order condition for $w$ implies that $\nabla f(w) + \lambda(w - z) = 0$, so $\lambda^2\|w - z\|^2 = \|\nabla f(w)\|^2$. Combining this with the Polyak-Łojasiewicz inequality, we obtain

$$\frac{\lambda}{2}\|w - z\|^2 = \frac{1}{2\lambda}\|\nabla f(w)\|^2 \geq \frac{m}{\lambda}(f(w) - f(x^*))$$

We have $\psi(w) \leq \psi(z) = f(z)$, so $f(w) - f(x^*) \leq f(z) - f(x^*) - \frac{\lambda}{2}\|w - z\|^2$. Substituting, $f(w) - f(x^*) \leq f(z) - f(x^*) - \frac{m}{\lambda}(f(w) - f(x^*))$, which yields the first inequality.

We also have $\psi(w) = f(w) + \frac{\lambda}{2}\|w - z\|^2 \leq \psi(x^*) = f(x^*) + \frac{\lambda}{2}\|x^* - z\|^2$, whence $f(w) - f(x^*) \leq \frac{\lambda}{2}(\|x^* - z\|^2 - \|w - z\|^2)$. Hence, we have

$$f(w) - f(x^*) \leq \frac{\lambda}{2}(\|x^* - z\|^2 - \|w - z\|^2)$$

$$\leq \frac{\lambda}{2}\|z - x^*\|^2 - \frac{m}{\lambda}(f(w) - f(x^*))$$

so $f(w) - f(x^*) \leq \frac{\lambda^2}{2(m+\lambda)}\|z - x^*\|^2$. Finally, by Theorem 10, $f(w) - f(x^*) \geq \frac{m}{2}\|w - x^*\|^2$, which yields the result. □

### 7.6 Proofs from Section 3.1.3

We first present two lemmas which consider the problem of selecting the minimizer from a collection, based on a single estimate of the value of each item.

**Lemma 16.** *Let $\mu_1 \leq \mu_2 \leq \ldots \leq \mu_p$. Suppose that $Y_1, \ldots, Y_p$ is a collection of random variables with $\mathbb{E}Y_i = \mu_i$ and $\mathrm{Var}(Y_i) \leq \sigma^2$. Let $\widetilde{\mu} = \mu_m$ where $m = \mathrm{argmin}\{Y_1, \ldots, Y_p\}$. Then*

$$\Pr(\widetilde{\mu} \geq \mu_k) \leq 4\sigma^2 \sum_{i=k}^{p} \frac{1}{(\mu_i - \mu_1)^2}$$

*Therefore, for any $a \geq 0$,*

$$\Pr(\widetilde{\mu} \geq \mu_1 + a) \leq 4\sigma^2 \frac{p}{a^2}.$$

*Proof.* In order for $\mu_m \geq \mu_k$, we must have $Y_j \leq Y_1$ for some $j \geq k$. Thus, $\{\widetilde{\mu} \geq \mu_k\}$ is a subset of the event $\{Y_1 \geq \min\{Y_k, \ldots, Y_p\}\}$. Taking the union bound,

$$\Pr(Y_1 \geq \min\{Y_k, \ldots, Y_p\}) \leq \sum_{i=k}^{p} \Pr(Y_1 \geq Y_i)$$

Applying Chebyshev's inequality to $Y_1 - Y_i$, and noting that $\operatorname{Var}(Y_1 - Y_i) \leq 4\sigma^2$ (if $Y_1, Y_i$ are independent, then this can be tightened to $2\sigma^2$), we have

$$\Pr(Y_1 - Y_i \geq 0) \leq \Pr(|Y_1 - Y_i - (\mu_i - \mu_1)| \geq \mu_i - \mu_1) \leq \frac{4\sigma^2}{(\mu_i - \mu_1)^2}.$$

$\square$

**Lemma 17.** *Let $\widetilde{\mu}$ be defined as in Theorem 16. Then*

$$\mathbb{E}\widetilde{\mu} - \mu_1 \leq 4\sqrt{p}\sigma$$

*Proof.* Recall that the expected value of a non-negative random variable $Z$ can be expressed as $\mathbb{E}Z = \int_0^\infty \Pr(Z \geq t)dt$. We apply this to the variable $\widetilde{\mu} - \mu_1$. Using Theorem 16, we obtain, for any $a > 0$,

$$\mathbb{E}\widetilde{\mu} - \mu_1 = \int_0^\infty \Pr(\widetilde{\mu} - \mu_1 \geq t)dt = \int_0^a \Pr(\widetilde{\mu} - \mu_1 \geq t)dt + \int_a^\infty \Pr(\mu^* - \mu_1 \geq t)dt$$

$$\leq a + \int_a^\infty \Pr(\widetilde{\mu} - \mu_1 \geq t)dt$$

$$\leq a + \int_a^\infty 4\sigma^2 \frac{p}{t^2}dt = a + 4\sigma^2 \frac{p}{a}$$

The AM-GM inequality implies that $a + 4\sigma^2 \frac{p}{a} \geq 4\sqrt{p}\sigma$, with equality when $a = 2\sqrt{p}\sigma$. $\square$

We now apply this to stochastic leader selection in LSGD, where $\mu_i$ corresponds to the true value $f(x^i)$, and $Y_i$ is a function estimator.

**Lemma 18.** *Let $f$ satisfy Assumption 1. Suppose that LSGD has a gradient estimator with $\operatorname{Var}(\widetilde{g}(x)) \leq \sigma^2 + \nu\|\nabla f(x)\|^2$ and selects the stochastic leader with a function estimator $\widetilde{f}(x)$ with $\operatorname{Var}(\widetilde{f}(x)) \leq \sigma_f^2$. Then, taking the expectation with respect to the gradient estimator and the stochastic leader $z$, we have*

$$\mathbb{E}f(x_+) \leq f(x) + 4\eta\lambda\sqrt{p}\sigma_f + \frac{\eta^2}{2}M\sigma^2$$

$$- \frac{\eta}{2}(1 - \eta M(\nu + 1))\|\nabla f(x)\|^2 - \frac{\eta}{4}\lambda(m - 2\eta M\lambda)\|\delta\|^2 - \frac{\eta\sqrt{\lambda}}{\sqrt{2}}(\sqrt{m} - \eta M\sqrt{2\lambda})\|\nabla f(x)\|\|\delta\|$$

*Proof.* From Theorem 12, we obtain

$$\mathbb{E}f(x_+) \leq f(x) - \frac{\eta}{2}(1 - \eta M(\nu + 1))\|\nabla f(x)\|^2$$

$$- \frac{\eta}{4}\lambda(m - 2\eta M\lambda)\|\delta\|^2$$

$$- \frac{\eta\sqrt{\lambda}}{\sqrt{2}}(\sqrt{m} - \eta M\sqrt{2\lambda})\|\nabla f(x)\|\|\delta\|$$

$$- \eta\lambda(f(x) - \mathbb{E}f(z)) + \frac{\eta^2}{2}M\sigma^2$$

Note that in the last line, we have $\mathbb{E}f(z)$ because $z$ is now stochastic. Applying Theorem 17 to the stochastic leader, we obtain $\mathbb{E}f(z) \leq f(z_{true}) + 4\sqrt{p}\sigma_f$. The true leader satisfies $f(z_{true}) \leq f(x)$ by definition. Hence $f(x) - \mathbb{E}f(z) \geq f(x) - f(z_{true}) - 4\sqrt{p}\sigma_f \geq -4\sqrt{p}\sigma_f$, and so $-\eta\lambda(f(x) - \mathbb{E}f(z)) \leq 4\eta\lambda\sqrt{p}\sigma_f$. $\square$

**Theorem 19.** *Let $f$ satisfy Assumption 1. If $\eta, \lambda$ are fixed so that $\eta \le (2M(\nu + 1))^{-1}$ and $\eta\lambda \le (2\kappa)^{-1}, \eta\sqrt{\lambda} \le (\kappa\sqrt{2m})^{-1}$, then $\limsup_{k\to\infty} \mathbb{E}f(x_k) - f(x^*) \le \frac{1}{2}\eta\kappa\sigma^2 + \frac{4}{m}\lambda\sqrt{p}\sigma_f$. If $\eta, \lambda$ decrease at the rate $\eta_k = \Theta(\frac{1}{k}), \lambda_k = \Theta(\frac{1}{k})$, then $\mathbb{E}f(x_k) - f(x^*) = O(\frac{1}{k})$.*

*Proof.* Interpret the term $4\eta\lambda\sqrt{p}\sigma_f$ as additive noise. Note that if $\eta_k, \lambda_k = \Theta(\frac{1}{k})$, then $\eta\lambda = \Theta(\frac{1}{k^2})$. The proof is then similar to Theorem 13 and follows from Theorems 4.6 and 4.7 of [30]. □

### 7.7 Proofs from Section 3.2

**Theorem 20.** *Let $\Omega_i$ be the set of points $(x^1, \ldots, x^p)$ where $x^i$ is the unique minimizer among $(x^1, \ldots, x^p)$[6]. Let $x^* = (w^1, \ldots, w^p) \in \Omega_i$ be a stationary point of the LGD objective function* (6). *Then $\nabla f^i(w^i) = 0$.*

*Proof.* This follows from the fact that on $\Omega_i$, $\frac{\partial L}{\partial x^i} = \nabla f^i(x^i)$. □

**Lemma 21.** *Let $f$ be $M$-Lipschitz-differentiable. Let $\widetilde{x}_k$ denote the leader at the end of the $k$-th period. If the LGD step size is chosen so that $\eta^i < \frac{2}{M}$, then $f(\widetilde{x}_k) \le f(\widetilde{x}_{k-1})$.*

*Proof.* Assume that $\widetilde{x}_{k-1} = x^1_{k-1}$. Since $x^1$ is the leader during the $k$-th period, the LGD steps for $x^1$ are gradient descent steps. By Theorem 11, $\eta^1$ has been chosen so that gradient descent on $f$ is monotonically decreasing, so we know that $f(x^1_k) \le f(x^1_{k-1})$. Hence $f(\widetilde{x}_k) \le f(x^1_k) \le f(x^1_{k-1}) = f(\widetilde{x}_{k-1})$. □

**Theorem 22.** *Assume that $f$ is bounded below and $M$-Lipschitz-differentiable, and that the LGD step sizes are selected so that $\eta^i < \frac{2}{M}$. Then for any choice of communication periods, it holds that for every $i$ such that $x^i$ is the leader infinitely often, $\liminf_k \|\nabla f(x^i_k)\| = 0$.*

*Note that there necessarily exists an index $i$ such that $x^i$ is the leader infinitely often.*

*Proof.* Without loss of generality, we assume it to be $x^1$. Let $\tau(1), \tau(2), \ldots$ denote the periods where $x^1$ is the leader, with $b(k)$ steps in the period $\tau(k)$. By Theorem 21, $f(x^1_{\tau(k+1)}) \le f(x^1_{\tau(k)})$, since the objective value of the leaders is monotonically decreasing. Now, by Theorem 11, we have $\sum_{i=0}^{b(k)-1} \|\nabla f(x^1_{\tau(k),i})\|^2 \le \alpha(f(x^1_{\tau(k),0}) - f(x^1_{\tau(k),b(k)})) = \alpha(f(x^1_{\tau(k)}) - f(x^1_{\tau(k+1)}))$. Since $f$ is bounded below, and the sequence $\{f(x^1_{\tau(k)})\}$ is monotonically decreasing, we must have $f(x^1_{\tau(k)}) - f(x^1_{\tau(k+1)}) \to 0$. Therefore, we must have $\|\nabla f(x^1_{\tau(k),i})\| \to 0$. □

### 7.8 Proofs from Section 3.3

The *cone* with center $d$ and angle $\theta_c$ is defined to be

$$\text{cone}(d, \theta_c) = \{x : x^T d \ge 0, \theta(x, d) \le \theta_c\}.$$

We record the following facts about cones which will be useful.

**Proposition 23.** *Let $C \subseteq \text{cone}(d, \theta_c)$. If $y$ is a point such that $sy \in C$ for some $s \ge 0$, then $y \in \text{cone}(d, \theta_c)$.*

*Proof.* This follows immediately from the fact that $\theta(y, d) = \theta(sy, d)$ for all $s \ge 0$. □

**Proposition 24.** *Let $C = \text{cone}(d, \theta_c)$ with $\theta_c > 0$. The outward normal vector at the point $x \in \partial C$ is given by $N_x = x - \frac{\|x\|}{\cos(\theta_c)\|d\|}d$. Moreover, if $v$ satisfies $N_x^T v < 0$, then for sufficiently small positive $\lambda$, $x + \lambda v \in \text{cone}(d, \theta_c)$.*

*Proof.* The first statement follows from the second, by the supporting hyperplane theorem.

Write $\gamma = \cos(\theta_c)$. Let $N_x = x - \frac{\|x\|}{\gamma\|d\|}d$, and let $v$ be a unit vector with $N_x^T v = x^T v - \frac{\|x\|}{\gamma\|d\|}d^T v < 0$. The angle satisfies

$$\cos(\theta(x + \lambda v, d)) = \frac{d^T(x + \lambda v)}{\|d\|\|x + \lambda v\|} = \frac{d^T x + \lambda d^T v}{\|d\|\sqrt{\|x\|^2 + \lambda^2\|v\|^2 + 2\lambda x^T v}}$$

Differentiating, the numerator $g(\lambda)$ of $\frac{\partial}{\partial\lambda}\cos(\theta(x + \lambda v, d))$ is given by

$$g(\lambda) = \|x\|^2 v^T d - x^T v x^T d + \lambda \cdot (2v^T d x^T d + \|v\|^2(\lambda v - x)^T d - \lambda\|v\|^2 v^T d - x^T v v^T d)$$

Evaluating at $\lambda = 0$ and using $x^T v - \frac{\|x\|}{\gamma\|d\|}d^T v < 0$, we obtain

$$g(0) = \|x\|^2 v^T d - x^T v x^T d = \|x\|^2 v^T d - x^T v(\gamma\|x\|\|d\|)$$
$$= \|x\|(\|x\|v^T d - \gamma\|d\|x^T v) > 0.$$

Therefore, for small positive $\lambda$, we have $\cos(\theta(x + \lambda v, d)) > \cos(\theta(x, d)) = \gamma$, so $x + \theta v \in \mathrm{cone}(d, \theta_c)$. □

**Proposition 25.** *Let $x$ be any point such that $\theta_x = \theta(d_G(x), d_N(x)) > 0$, and let $E = \{z : f(z) \leq f(x)\}$. Let $C = \mathrm{cone}(-x, \theta_x)$, and let $N_x$ be the outward normal $-\nabla f(x) + \frac{\|\nabla f(x)\|}{\cos(\theta_x)\|x\|}x$ of the cone $C$ at the point $-\nabla f(x)$. Then*

$$\bigcup_{\lambda > 0} I_\theta(x, \lambda) \supseteq E \cap \{z : N_x^T z < N_x^T x\} \tag{14}$$

*and consequently, $\lim_{\lambda \to 0} \mathrm{Vol}(I_\theta(x, \lambda)) \geq \frac{1}{2}\mathrm{Vol}(E)$.*

*Proof.* First, note that if $\lambda_2 \leq \lambda_1$, then for all $z$ with $-\nabla f(x) + \lambda_1 z \in C$, we also have $-\nabla f(x) + \lambda_2 z \in C$ by the convexity of $C$. Therefore $I_\theta(x, \lambda_2) \supseteq I_\theta(x, \lambda_1)$, so $\lim_{\lambda \to 0}\mathrm{Vol}(I_\theta(x, \lambda))$ exists. We first prove the second statement. For any normal vector $h$ and $\beta > 0$, $\mathrm{Vol}(E \cap \{z : h^T z < \beta\}) \geq \frac{1}{2}\mathrm{Vol}(E)$, since the center $0 \in \{z : h^T z < \beta\}$. The result follows because $N_x^T x > 0$.

To prove (14), observe that $z \in I_\theta(x, \lambda)$ if equivalent to $-\nabla f(x) + \lambda(z - x) \in \mathrm{cone}(-x, \theta_c)$. By Theorem 24, there exists $\lambda > 0$ with $-\nabla f(x) + \lambda(z - x) \in \mathrm{cone}(-x, \theta_c)$ if $N_x^T(z - x) < 0$. Hence, it follows that every point in $E \cap \{z : N^T z < N^T x\}$ is contained in $I_\theta(x, \lambda)$ for some $\lambda > 0$. □

**Lemma 26.** *There exists a direction $x$ such that $\cos(\theta(d_G(x), d_N(x))) = 2(\sqrt{\kappa} + \sqrt{\kappa^{-1}})^{-1}$. Thus, for all $r \geq 2$, there exists a direction $x$ with $\cos(\theta(d_G(x), d_N(x))) \leq \frac{r}{\sqrt{\kappa}}$.*

*Proof.* Take $x = \sqrt{\frac{\alpha_n}{\alpha_1 + \alpha_n}}e_1 + \sqrt{\frac{\alpha_1}{\alpha_1 + \alpha_n}}e_n$. It is easy to verify that $\cos(\theta(d_G, d_N)) = 2(\sqrt{\kappa} + \sqrt{\kappa^{-1}})^{-1}$. □

**Proposition 27.** *For any $x$, let $\theta_x = \theta(d_G(x), d_N(x))$. We have*

$$\max\{\|z\|^2 : f(z) \leq f(x), z^T x = 0\} \leq \kappa \cos(\theta_x)\|x\|^2$$

*Proof.* Form the maximization problem

$$\begin{cases} \max\limits_z & z^T z \\ & z^T A z \leq x^T A x \\ & z^T x = 0 \end{cases}$$

The KKT conditions for this problem imply that the solution satisfies $z - \mu_1 A z - \mu_2 x = 0$, for Lagrange multipliers $\mu_1 \geq 0, \mu_2$. Since $z^T x = 0$, we obtain $z^T z = \mu_1 z^T A z$, and thus $\frac{1}{M} \leq \mu_1 \leq \frac{1}{m}$. Since $f(z) \leq f(x)$, we find that $z^T z \leq \frac{1}{m}x^T A x$. Using $\cos(\theta_x) = \frac{x^T A x}{\|x\|\|Ax\|}$, we obtain

$$z^T z \leq \frac{1}{m}\cos(\theta_x)\|x\|\|Ax\| \leq \kappa \cos(\theta_x)\|x\|^2.$$

□

**Theorem 28.** *Let $R_\kappa = \{r : \frac{r}{\sqrt{\kappa}} + \frac{r^{3/2}}{\kappa^{1/4}} \leq 1\}$. Let $x \in S_r$ for $r \in R_\kappa$, and let $E = \{y : f(y) \leq f(x)\}$, $E_2 = \{z \in E : z^T x \leq 0\}$, $\theta_x = \theta(d_G(x), d_N(x))$. Then for all $z \in E_2$ and any $\lambda \geq 0$, the LGD direction $d_z = -(\nabla f(x) + \lambda(x - z))$ satisfies $\theta(d_z, d_N(x)) \leq \theta_x$. Thus, $E_2 \subseteq I_\theta(x, \lambda)$, and therefore $\text{Vol}(I_\theta(x, \lambda)) \geq \text{Vol}(E_2) = \frac{1}{2}\text{Vol}(E)$.*

*Proof.* Define $D_2 = \{z - x : z \in E_2\}$[7]. The set of possible LGD directions with $z \in E_2$ is given by $D_3 = \{-\nabla f(x) + \lambda\delta : \delta \in D_2, \lambda \geq 0\}$. Since $d_N(x) = -x$, our desired result is equivalent to $D_3 \subseteq \text{cone}(-x, \theta_x)$.

Define the subset $D_2' = \{z - x : z \in E_2, x^T z = 0\}$. We claim that it suffices to prove that $D_2' \subseteq \text{cone}(-x, \theta_x)$. To see this, consider any $\lambda\delta$ for $\lambda \geq 0$ and $\delta \in D_2$. We have $x^T(\lambda\delta) = \lambda x^T(z - x) \leq -\lambda x^T x < 0$, so there exists a scalar $s$ with $x^T(s\lambda\delta) = -x^T x$, whence $s\lambda\delta \in D_2' \subseteq \text{cone}(-x, \theta_x)$. By Theorem 23, $\lambda\delta \in \text{cone}(-x, \theta_x)$. Since $-\nabla f(x) \in \text{cone}(-x, \theta_x)$, convexity implies that $-\nabla f(x) + \lambda\delta \in \text{cone}(-x, \theta_x)$. Thus, $D_2' \subseteq \text{cone}(-x, \theta_x)$ implies that $D_3 \subseteq \text{cone}(-x, \theta_x)$.

To complete the proof, let $\delta = z - x \in D_2'$ and observe that $\cos(\theta(\delta, d_N(x))) = \frac{x^T(x-z)}{\|x\|\|x-z\|}$. By Theorem 27 and the definition of $S_r$,

$$\max\{\|z\| : z \in E_2, z^T x = 0\} \leq \sqrt{\kappa}\sqrt{\cos(\theta_x)}\|x\| = \sqrt{r}\kappa^{1/4}\|x\|$$

We compute that

$$x^T(x - z) - \frac{r}{\sqrt{\kappa}}\|x\|\|x - z\| \geq \|x\|^2 - \frac{r}{\sqrt{\kappa}}(\|x\|^2 + \|x\|\|z\|)$$

$$\geq \|x\|^2 - \frac{r}{\sqrt{\kappa}}\|x\|^2 - \frac{r}{\sqrt{\kappa}}\|x\|(\sqrt{r}\kappa^{1/4}\|x\|)$$

$$\geq \left(1 - \frac{r}{\sqrt{\kappa}} - \frac{r^{3/2}}{\kappa^{1/4}}\right)\|x\|^2 \geq 0$$

By the definition of $R_\kappa$, this is non-negative, and thus $\theta(\delta, d_N(x)) \leq \theta_x$. This completes the proof. $\qquad\square$

# 8 Low-Rank Matrix Completion Experiments

Low-rank matrix completion problem is an example of a non-convex learning problem whose landscape exhibits numerous symmetries. We consider the positive semi-definite case, where the objective is to find a low-rank matrix minimizing

$$\min_X \left\{ F(X) = \frac{1}{4}\|M - XX^T\|_F^2 : X \in \mathbb{R}^{d \times r} \right\}$$

It is routine to calculate that $\nabla F(X) = (XX^T - M)X$. The EAGD and LGD updates for $X$ can be expressed as

$$X_+ = (1 - \eta\lambda)X + \eta\lambda Z - \eta\nabla F(X).$$

For EAGD, $Z = \widetilde{X}$, and $\widetilde{X}$ is updated by

$$\widetilde{X}_+ = (1 - p\eta\lambda)\widetilde{X} + p\eta\lambda\left(\frac{1}{p}\sum_{i=1}^{p} X^i\right).$$

For LGD, $Z = \arg\min\{F(X^1), \ldots, F(X^p)\}$, and is updated at the beginning of every communication period $\tau$.

The parameters were set to:

$$\eta = \texttt{5e-4}, \lambda = \frac{1}{5}, p = 8, \tau = 1$$

The learning rate $\eta = \texttt{5e-4}$ was selected from a set $\{\texttt{1e-1}, \texttt{5e-2}, \texttt{1e-3}, \ldots\}$ by evaluating on a sample problem until a value was found for which both methods exhibited monotonic decrease.

The dimension was $d = 1000$, and the ranks $r \in \{1, 10, 50, 100\}$ were tested. For each rank, there were 10 random trials performed. In each trial, $M$ and starting points $\{X_0^i\}$ are sampled. $M$ is generated by sampling $U \in \mathbb{R}^{d \times r}$ with i.i.d entries from $N(0, 1)$, and taking $M = UU^T$. Initial points for each worker node $X^i$ were also sampled from $N(0, 1)$. The same starting points were used for EAGD and LGD.

Code for this experiment is available at `https://github.com/wgao-res/lsgd_matrix_completion`.

# 9 Experimental Setup

## 9.1 Data preprocessing

For CIFAR-10 experiments we use the original images of size $3 \times 32 \times 32$. We then normalize each image by mean $(0.4914, 0.4822, 0.4465)$ and standard deviation $(0.2023, 0.1994, 0.2010)$. We also augment the training data by horizontal flips with a probability of $0.5$.

For CNN7 and ResNet20, we extract random crops of size $3 \times 28 \times 28$ and present these to the network in batches of size 128. The test loss and test error are only computed from the center patch $(3 \times 28 \times 28)$ of test images.

For VGG16 we pad the images to $3 \times 40 \times 40$, extract random crops of size $3 \times 32 \times 32$ and present these to the network in batches of size 128. The test loss and test error are computed from the test images.

For ImageNet experiments we normalize each image by mean $(0.485, 0.456, 0.406)$ and standard deviation $(0.229, 0.224, 0.225)$. We sample the training data in the same way as [39]. For each image, a crop of random size (chosen from $8\%$ to $100\%$ evenly) of the original size and a random aspect ratio (chosen from $3/4$ to $4/3$ evenly) of the original aspect ratio is made. Then we resize the crop to $3 \times 224 \times 224$. We also augment the training data by horizontal flips with a probability of $0.5$. Finally we present these to the network in the batches of size 32. The test images are resized so that the smaller edge of each image is 256. The test loss and test error are only computed from the center patch $(3 \times 224 \times 224)$ of test images.

## 9.2 Data prefetching

We use the dataloader and distributed data sampler[8] from PyTorch. Each worker loads a subset of the original data set that is exclusive to that worker for every epoch. If the size of data set is not divisible by the batch size, the last incomplete batch will be dropped.

## 9.3 Hyperparameters

In Table 2 we summarize the learning rates and other hyperparameters explored for each method in the CNN7 experiment on CIFAR-10. The setting of $\beta$ for EASGD was obtained from the original paper (its authors use this setting for all their experiments).

Table 2: Hyperparameters: CNN7 experiment on CIFAR-10

| Name | Learning Rates | |
|---|---|---|
| SGD | $\{0.1, 0.05, 0.01, 0.005, 0.001\}$ | |
| DOWNPOUR | $\{0.05, 0.01, 0.005, 0.001, 0.0005\}$ | |
| EASGD | $\{0.1, 0.05, 0.01, 0.005, 0.001\}$ | $\beta = 0.43$ |
| LSGD | $\{0.1, 0.05, 0.01, 0.005, 0.001\}$ | $\lambda = \{0.5, 0.2, 0.1, 0.05, 0.025\}, \lambda_G = \lambda$ |

In Table 3 we summarize the initial learning rates and other hyperparameters explored for each method in the ResNet20 experiment on CIFAR-10. We do learning rate drop at 1500 seconds by a factor of $0.1$ for all the methods.

Table 3: Hyperparameters: ResNet20 experiment on CIFAR-10

| Name | Learning Rates | |
|---|---|---|
| SGD | $\{0.2, 0.1, 0.05\}$ | |
| DOWNPOUR | $\{0.2, 0.1, 0.05, 0.01\}$ | |
| EASGD | $\{0.2, 0.1, 0.05\}$ | $\beta = 0.43$ |
| LSGD | $\{0.2, 0.1, 0.05\}$ | $\lambda = \{0.5, 0.2, 0.1, 0.05, 0.025\}, \lambda_G = \lambda$ |

In Table 4 we summarize the learning rates and other hyperparameters explored for each method in the VGG16 experiment on CIFAR-10. We do learning rate drop at 1500 seconds by a factor of 0.1 for all the methods.

Table 4: Hyperparameters: VGG16 experiment on CIFAR-10

| Name | Learning Rates | |
|---|---|---|
| SGD | $\{0.2, 0.1, 0.05\}$ | |
| DOWNPOUR | $\{0.2, 0.1, 0.05, 0.01\}$ | |
| EASGD | $\{0.2, 0.1, 0.05\}$ | $\beta = 0.43$ |
| LSGD | $\{0.2, 0.1, 0.05\}$ | $\lambda = \{0.2, 0.1\}$ |

In Table 5 we summarize the initial learning rates and other hyperparameters explored for each method in the ResNet50 experiment on ImageNet. We do learning rate drop for every 30 epochs by a factor of 0.1 for all the methods.

Table 5: Hyperparameters: ResNet50 experiment on ImageNet

| Name | Learning Rate | |
|---|---|---|
| DOWNPOUR | 0.1 | |
| EASGD | 0.2 | $\beta = 0.43$ |
| LSGD | 0.2 | $\lambda = 0.1$ |

### 9.4 Implementation Details

To take advantage of both the efficiency of collective communication and the flexibility of peer-to-peer communication, we incorporate two backends, namely NCCL and GLOO[9], for GPU processors and CPU processors, respectively.

The global and local servers (running on CPU processors) control the training process and the workers (running on GPU processors) perform the actual computations. For each iteration each worker has only one of the following two choices:

1. Local Training: Each worker is trained with one batch of the training data;

2. Distributed Training: Each worker communicates with other workers and updates its parameters based on the pre-defined distributed training method.

To minimize the cost of communication over Ethernet, the global server is running on the first GPU node instead of a separate machine. Also, for a fair comparison, the center variable is being maintained and updated by the first GPU node as well[10].

Figure 8: At the beginning of each iteration, the local worker sends out a request to its local server and then the local server passes on the worker's request to the global server. The global server checks the current status and replies to the local server. The local server passes on the global server's message to the worker. Finally, depending on the message from the global server, the worker will choose to follow the local training or distributed training scheme.

## 10 Additional Experimental Results

### 10.1 Word-level Language Model

We train an LSTM model on Wikitext-2 for word-level text prediction. Our network consists of two layers with 200 hidden units and we set the sequence length to 35. The implementation is adapted from PyTorch example[11]. In Figure 9 we show that LSGD outperforms other comparators.

Figure 9: LSTM on Wikitext-2. Test perplexity for the center variable versus wall-clock time. The number of workers is set to 4.

### 10.2 More results from Section 4.2

Figure 10: CNN7 on CIFAR-10. Test loss for the center variable versus wall-clock time (original plot on the left and zoomed on the right).

Figure 11: ResNet20 on CIFAR-10. Test loss for the center variable versus wall-clock time (original plot on the left and zoomed on the right).

Figure 12: VGG16 on CIFAR-10. Test loss for the center variable versus wall-clock time (original plot on the left and zoomed on the right).

Figure 13: ResNet50 on ImageNet. Test loss for the center variable versus wall-clock time (original plot on the left and zoomed on the right).

## 11 Communication Efficiency

We report the proportion of communication costs with respect to the total time in Table 6. LSGD is roughly twice more communication-efficient than EASGD. Note that EASGD and DOWNPOUR require more time for data transmission and computation during communication as parameter updates involve an additional center variable.

Table 6: Proportion of communication costs with repect to the total time. Communication cost includes both data transmission and computation.

|  | LSGD | EASGD | DOWNPOUR |
|---|---|---|---|
| CNN7: 4/16 workers | 1%/2% | 2%/4% | 20%/57% |
| ResNet20: 4/16 workers | 1%/2% | 2%/4% | 21%/50% |
| VGG16 | 2% | 3% | 34% |
| ResNet50 | 1% | 2% | 17% |

## Footnotes

[5]If we also assume that $f$ is Lipschitz-differentiable (that is, $\nabla^2 f(x) \preceq MI$), then we can obtain a similar inequality to the second directly from the first, but this is generally weaker than the bound given here.

[6]The uniqueness of the minimizer on $\Omega_i$ is only to avoid ambiguities in $\arg\min$.

[7]Note the sign change from $x - z$ to $z - x$ here.

[8]https://pytorch.org/docs/stable/data.html

[9]https://github.com/facebookincubator/gloo

[10]In the original implementation of [1] and [9], an individual parameter server is used for updating the center variable based on the peer-to-peer communication scheme. However, there is no need to use an individual parameter server under collective communication scheme as it will only induce extra communication cost.

[11]https://github.com/pytorch/examples/tree/master/word_language_model