[Reviews · NeurIPS 2019]

Reviewer 1



Without rehashing the "contributions," the authors have done a great job of justifying their approach theoretically, and I enjoyed reading the paper. It is very clearly written, and the approach is well introduced. There are a couple small concerns. First, the paper is extremely dependent upon the supplemental material, and it's difficult to sufficiently understand the approach without reading it. Second, the experiments are not actually that overwhelming in practical performance; they are only slightly better or merely comparable to other approaches. Finally, I don't think "Vol" is defined in Theorem 6, so I don't know what's going on there. There is a great deal of intuitive justification for the approach based on the idea that averaging between workers can result in pulling the worker out of a minimum. This seems just as likely to happen to the non-leader workers in this approach. Is this indicated in the schedule of the leader switching? Does it tend to stick with a single leader for a long time, while the other workers are stranded in averaged locations?

Reviewer 2



This paper proposed a new algorithm called LSGD for distributed optimization in non-convex settings based on pulling workers to the current best performer among them, rather than their average at each iteration. Originality: using the leader stochastic gradient descent is interesting. Quality: the theoretical analysis is rigorous. Clarity: the paper is easy to understand including the motivating example, and the problem formulation. They do not compare the convergence rate with existing methods theoretically, which is important to see the theoretical improvement.

Reviewer 3



After rebuttal: I have carefully read the authors' response. Unfortunately, I do not think my concerns are well addressed. (1) intuitively, i cannot see why many candidate leaders will be in directions of negative curvature; (2) there is neither theoretical nor empirical evidence that x_i's of all workers will lie in the same valley; (3) the test perplexity of additional experiment with LSTM on the NLP task looks very bad. See Table 2 in "Regularizing and Optimizing LSTM Language Models" for comparison; (4) the performance of SGD on a single GTX1080 GPU does not tell how it performs with multiple workers (larger mini-batch size); (5) selecting learning rate based on the test error is not a good practice. For machine learning, we should select the hyper-parameters according to the accuracy on a hold-out validation set. Considering the above five points, I decide to keep my score unchanged. This paper considers distributed training for nonconvex models such deep neural networks. Different from EASGD, the proposed leader gradient descent (LGD) replaces the averaged parameters with the parameters on "leader" worker, which gives the smallest objective value among all the workers. To further reduces the communication cost, the leader worker is updated every finite number of iterations, and asynchronous execution is considered. As the evaluation of the full objective is expensive, the authors proposed a stochastic extension, leader stochastic gradient descent (LSGD), in which the leader is selected based on the estimated value on a subsampled batch. The theoretical results show: 1) using guiding point improves one-step SGD and has an asymptotic O(1/t) rate for strongly convex problems; 2) the stationary points of the perturbed objective are the stationary points of the original objective; 3) leader selection makes the update direction more close Newton direction. The experiments on image classification task show that the proposed method outperforms EASGD. Overall, this is an interesting paper. However, I raise some questions, which need to be addressed by the authors, as follows: 1. The theoretical results only apply to the deterministic and simplified version of the proposed asynchronous LSGD algorithm. This makes it difficult to understand the convergence behaviour of the actual algorithm used in the experiment. For instance, as shown in Theorem 3, the stochastic leader selection contributes an extra error term and the convergence is not guaranteed unless the importance of the leader decreases over time. The LSGD method also behaves like local SGD method, and number of rounds for synchronization plays a significant role in the convergence. However, there is no analysis on how this hyper-parameter affects the convergence rate of the proposed approach. 2. In Section 3.3, it is shown that at least half of the candidate leaders result in a search direction that is closer to Newton direction. This is a good result for convex optimization. However, as the focus of the paper is on nonconvex problems and it has been shown that Newton iterates are attracted to saddle points (Dauphin et al., 2014), the claimed statement is not convincing in the context of the training of the deep neural networks. 3. The authors state that the averaging step hurts the convergence, which is true in the symmetric non-convex landscapes. However, there are some cases that the averaged iterate improve the performance, such as training AWD-LSTM model for language modeling and Transformer network for translation task. Thus, it is also good to conduct experiments on such NLP tasks to see how the LSGD method behaves. Besides, as claimed in the paper that the averaging step hurts the performance, then why don't you use the leader's parameter instead of the average of the parameters of all the workers, which is used in the experiment, to perform the testing? 4. SGD is compared on all the CIFAR-10 experiments. However, SGD is not compared on training the ResNet50 on ImageNet dataset. I guess that SGD can perform significantly better than LSGD in this case. I am also curious about how the proposed method performs on a deeper ResNet (e.g., ResNet110) on the CIFAR-10 dataset. 5. There is no description on how the best learning rate is selected for each method in the supplementary. 6. As the authors claim that LSGD is more communication-efficient, it is necessary to also report the breakdown of the total time into computation and communication costs. Minor comments: 1. In Theorem 4, as x^i has been previously defined, it would be better to use different symbol. 2. The definition of f^i is missing in Theorem 4. 3. There are many typos in the paper/supplementary. Dauphin et al., 2014. Identifying and attacking the saddle point problem in high-dimensional non-convex optimization. NIPS 2014.

[Author Response · NeurIPS 2019]

We thank all the Reviewers for their time and raising several interesting questions. We appreciate kind comments.
Please see our responses below.

Reviewer #1: We will try to reduce dependence on the Supplement. Regarding the non-leader workers in LSGD,
we believe that these two facts together —- (1) the schedule of leader switching recorded in the experiments shows
frequent switching, and (2) the leader point itself is not pulled away from minima — suggest that the 'pulling away' in
LSGD is beneficial: non-leader workers that were pulled away from local minima later became the leader, and thus
likely obtained an even better solution than they originally would have. We will add an explanation of this interesting
phenomenon in the final paper. The name Vol in §3.3 refers to Volume, which for the ellipsoid $\{x : x^T A x \leq 1\}$ is
given by $\det(A)^{-1/2} \text{Volume}(S_n)$, where $S_n$ is the unit ball. We will add this definition.

Reviewer #2: We will add a comment comparing the convergence rate of LSGD to other distributed methods. For
the DOWNPOUR method specifically, the original paper is purely empirical, and we are not aware of any published
convergence analysis for it. The closest proxy for DOWNPOUR with a known rate is the Hogwild method of Recht, B.
et al. [2011], which achieves a rate of $O(1/k)$ (up to a logarithmic term). The EASGD method also achieves $O(1/k)$
on strongly convex objective functions. In both cases, this rate matches that of LSGD. Unfortunately many distributed
algorithms are presented without theoretical analysis (e.g. DOWNPOUR, PARLE).

Reviewer #3: Regarding the asynchronous algorithm, we opted to present results for two settings, one-step round and
'arbitrarily long' round, in the main paper because numerous variations of communication schedules are possible. The
behavior of the algorithm given an unknown round length >1 is very difficult to measure in a useful way (i.e., to find
quantifiable improvements) since it requires estimating the lowest value obtained along the trajectory when the leader is
kept fixed, which makes it difficult to define a rate for the 'general' asynchronous method. Note that several useful
lemmas for the analysis of stochastic leader selection are presented in the Supplement, which can also be combined to
analyze combinations of the asynchronous algorithm with stochastic leader selection.

Regarding the similarity to Newton directions, note that our volume theorems apply only for convex functions (or in the
neighborhoods of local minimizers). They do not apply in general for nonconvex functions, and thus they do not imply
improvements or, conversely, harmful behavior. For nonconvex functions, our intuition is that many candidate leaders
will be in directions of negative curvature which would actually draw us away from saddle points, but measuring this
volume is significantly more complicated because the set of candidate leaders is a priori unbounded.

LSTM on Wikitext-2

Regarding using the leader's parameter versus the average of the parame-
ters of all the workers, note that we use the leader's parameter to pull to at
training and we report the averaged parameters at testing deliberately. It is
demonstrated in our paper (e.g.: Fig. 1) that pulling workers to the averaged
parameters at training may slow down convergence and we address this
problem. Note that after training, the parameters that workers obtained after
convergence will likely lie in the same valley of the landscape (see Baldassi,
C. et al. [2016]) and thus their average is expected to have better generaliza-
tion ability (e.g. Chaudhari P. et al. [2017], Izmailov, P. et al. [2018]), which
is why we report the results for averaged parameters at testing.

We report additional experiment with LSTM on the NLP task (world-level language modeling) in the figure above. We
are also running additional experiments. We will add them to the paper.

Regarding SGD and ResNet50, SGD is consistently worse than all reported methods (training on ImageNet with SGD
on a single GTX1080 GPU until convergence usually takes about a week and gives slightly worse final performance),
which is why the SGD curve was deliberately omitted (other methods converge in around two days).

|  | LSGD | EASGD | DOWNPOUR |
|---|---|---|---|
| CNN7: 4/16 workers | 1%/2% | 2%/4% | 20%/57% |
| ResNet20: 4/16 workers | 1%/2% | 2%/4% | 21%/50% |
| VGG16 | 2% | 3% | 34% |
| ResNet50 | 1% | 2% | 17% |

Regarding learning rate selection for each method, we
chose the one leading to the smallest achievable test error
under similar convergence rates (we rejected small learning
rates which led to unreasonably slow convergence).

The breakdown of the total time will be added, see provided
table reporting the proportion of communication costs with respect to the total time. LSGD is roughly twice more
communication-efficient than EASGD. Local SGD is indeed effective in terms of communication, but at the cost of a
significant drop in performance compared to DOWNPOUR, which is why we have not been reporting this method.

## References

Baldassi, C. et al. Unreasonable effectiveness of learning neural networks: From accessible states and robust ensembles to basic
algorithmic schemes. In *PNAS*, 2016.
Chaudhari P. et al. Entropy-SGD: Biasing gradient descent into wide valleys. In *ICLR*, 2017.
Izmailov, P. et al. Averaging weights leads to wider optima and better generalization. *arXiv:1803.05407*, 2018.
Recht, B. et al. Hogwild: A lock-free approach to parallelizing stochastic gradient descent. In *NIPS*, 2011.


[Meta-Review · NeurIPS 2019]

The paper proposes and theoretically analyzes a distributed SGD algorithm where the workers are pulled towards the best performing worker rather than the average worker. All three reviewers consider the theoretical contribution (analysis of convergence and cost of communication) to be interesting and rigorous. At the same time, one reviewer feels the theoretical analysis applies to a simplified case and may not shed light on the experiments that are done in more complex settings. The reviewer's were not satisfied by the rebuttal, but maintained that the paper is publishable. Overall, there is a consensus that is is a fine paper and I recommend acceptance.